# Utilization of Rehabilitation Services and Rehabilitation-Related Patient Satisfaction Following Total Knee Arthroplasty—Results of the Prospective FInGK Study

**DOI:** 10.3390/healthcare12212099

**Published:** 2024-10-22

**Authors:** Julius Oltmanns, Hannes Jacobs, Uwe Maus, Max Ettinger, Falk Hoffmann, Gesine H. Seeber

**Affiliations:** 1Department of Health Services Research, Carl-von-Ossietzky University Oldenburg, 26129 Oldenburg, Germany; hannes.jacobs@uni-oldenburg.de (H.J.); falk.hoffmann@uni-oldenburg.de (F.H.); 2University Hospital for Neurosurgery Evangelisches Krankenhaus Oldenburg, 26122 Oldenburg, Germany; 3Department of Orthopedic & Trauma Surgery, University Hospital of Düsseldorf, 40225 Düsseldorf, Germany; uwe.maus@med.uni-duesseldorf.de; 4Division of Orthopedics at Campus Pius-Hospital, School of Medicine and Health Sciences, Carl von Ossietzky Universität Oldenburg, 26121 Oldenburg, Germany; max.ettinger@uol.de (M.E.); or g.h.seeber@umcg.nl (G.H.S.); 5Department of Orthopedics, University Medical Center Groningen, University of Groningen, 9713 GZ Groningen, The Netherlands

**Keywords:** arthroplasty, rehabilitation, patient satisfaction

## Abstract

(1) Background: This study aims to examine rehabilitation service utilization among total knee arthroplasty (TKA) subjects and the influencing factors associated with rehabilitation-related satisfaction. (2) Methods: The FInGK study was a single-center prospective cohort study. Patients (≥18 years) undergoing primary or revision TKA in a German university hospital were consecutively recruited between December 2019–May 2021. The subjects filled in a questionnaire one day before surgery (t0) and at two (t1) and 12 (t2) months postoperatively. Multivariable logistic regression was conducted to determine the variables associated with the subjects’ rehabilitation-related satisfaction. (3) Results: A total of 236 out of 241 (97.9%) subjects participated in t1 (59.3% female; mean age: 68.2 years). Overall, 94.7% underwent post-TKA rehabilitation measures, with inpatient rehabilitation being the predominant choice (85.4%). In total, 77.6% of those with rehabilitation were satisfied or very satisfied with their rehabilitation in general. Multivariable logistic regression showed that female sex (OR 3.42; CI 1.73–6.75) and satisfaction with the surgery in general after two months (OR 4.50; CI 1.96–10.33) were associated with the subjects’ rehabilitation-related satisfaction. (4) Conclusions: We found a high utilization of rehabilitation services following TKA and a high rehabilitation-related satisfaction. In international comparison, the utilization of inpatient rehabilitation services was very high. Future research should investigate the effective components for rehabilitation-related satisfaction in both in- and outpatient TKA rehabilitation.

## 1. Introduction

Osteoarthritis is one of the most prevalent chronic articular disorders, leading to significant costs to the public healthcare system worldwide [1]. Notably, the knee joint is the area most frequently affected [2]. Given its chronic and progressive nature, symptoms tend to exacerbate over time, potentially causing a decline in the individual’s quality of life (QoL) [1]. Initial disease management adopts a conservative approach, focusing on symptom relief. Total knee arthroplasty (TKA), which ranks among the most frequently performed surgical interventions in developed nations, emerges as an effective management option when the limits of conservative therapeutic measures are reached [3,4]. Nonetheless, despite technological and surgical advancements, approximately 20% of patients remain dissatisfied with post-TKA outcomes. Complaints mainly center around a limited range of motion and persisting pain [5]. Therefore, it is assumed that the utilization of rehabilitation measures and their delivery significantly influence QoL after surgery [6].

The rehabilitation phase starts immediately after surgery. The earlier rehabilitation commences, the sooner the patient can achieve independent mobility, leading to a quicker recovery of walking with effective pain control [7]. The overarching goal during TKA rehabilitation is to alleviate residual osteoarthritis-related symptoms, such as persisting or postoperative pain and functional limitations of the knee joint, while maintaining or restoring QoL [8]. Key aspects to address during rehabilitation encompass patient education and weight loss, as well as improvements in joint motion, functional capacity, joint stability, and muscular coordination [9]. Cost-effective, and generally effective, approaches include physical modalities, exercise therapy, physiotherapy, and relaxation techniques [10,11,12,13].

Internationally, diverse strategies exist for structured post-TKA rehabilitation. In most countries, patients are discharged rapidly after surgery and proceed to outpatient rehabilitation. For instance, in England, inpatient rehabilitation is highly uncommon [14,15]. Similarly, most patients in the USA and Canada, undergo outpatient physical therapy during TKA rehabilitation [16]. The varying healthcare systems in each country are shaped by the different systems that have developed over time, thus also leading to different rehabilitation strategies. In Germany, for example, the healthcare system follows the Bismarck model of social insurance, which mandates health insurance and operates on an income-based system, with reimbursements being drawn from insurance contributions. In contrast, the USA has traditionally had minimal government involvement, and there was no mandatory health insurance until the Obamacare reform in 2014. In England, healthcare costs are covered by taxpayer funds under the principles of a welfare state [17]. In Germany, however, rehabilitation services are predominantly conducted in an inpatient setting [18]. Rehabilitation measures in a specialized rehabilitation clinic follow a multi-modal, multi-professional treatment approach [19], regularly commence within 14 days post-hospitalization, and are covered under the statutory pension insurance’s criteria [20]. Obtaining a rehabilitation prescription is part of standard TKA aftercare. Pension insurance covers working-age patient costs; for retirees, health insurance covers expenses [21].

Currently, the understanding of TKA rehabilitation as it is organized in Germany and rehabilitation-related patient satisfaction is limited. According to the 2021 German Pension Insurance Rehabilitation Report, 80% of all medical rehabilitations were inpatient, with musculoskeletal conditions accounting for most of the interventions. Female patients used rehabilitation services slightly more often versus male patients. Medical rehabilitation duration ranged from 22 to 24 days [18]. A study involving patients after total hip arthroplasty (THA) reported that 72% of patients underwent inpatient rehabilitation [22]. However, the data about TKA and rehabilitation-related patient satisfaction were lacking. No study to date has explicitly addressed the utilization of rehabilitation services and rehabilitation-related patient satisfaction post-TKA. The primary aim of this study was to provide comprehensive insights into rehabilitation service utilization and patient rehabilitation-related satisfaction following TKA.

## 2. Materials and Methods

### 2.1. Study Design, Study Population, and Sample

This single-center prospective cohort study (FInGK Study) was conducted at a specialized University Hospital for Orthopedics and Trauma Surgery in northwestern Germany. All of the adult patients who underwent elective primary and/or revision TKA between December 2019 and May 2021 were consecutively included. The exclusion criteria comprised (1) an age of ≤ 18 years, (2) a history of malignant neoplastic diseases with a life expectancy of less than 12 months, (3) insufficient German language proficiency or intellectual barriers that prevent the subjects from independently completing the questionnaire, and (4) patients who had already participated with TKA on the contralateral side. In-hospital rehabilitation measures were the same for all participants. The FInGK study aimed to investigate the subject-reported outcomes and the utilization of healthcare services before and after TKA [23]. Based on this, the a priori sample size calculation determined a minimum of 240 subjects [24]. All of the subjects provided written informed consent prior to enrollment. The local University’s Medical Ethics Committee a priori approved this study (#2019-064).

### 2.2. Data Collection and Information

Data were collected through a self-administered questionnaire. The subjects filled in the questionnaire one day before surgery (t0), as well as at two (t1) and 12 (t2) months postoperatively. For the current study’s analysis, the data from baseline (t0) and first follow-up at two months post-TKA (t1) were utilized. During both measurement timepoints, the subjects answered questions based on the following domains: (a) quality of life and health status; (b) pain, function, and well-being; (c) utilization of healthcare services; and (d) sociodemographic data and lifestyle factors. Moreover, the t1 questionnaire included additional questions regarding the subjects’ rehabilitation, satisfaction with the surgery, and subsequent rehabilitation, as well as the fulfillment of expectations. The length of hospital stay was extracted from the electronic medical record.

### 2.3. Rehabilitation Outcome Measurements

At t1, this study assessed whether the subjects had received a rehabilitation measure following their acute hospital stay, in what way rehabilitation was delivered (e.g., inpatient or outpatient), and the time interval between hospital discharge and the start of the rehabilitation measure. 

Rehabilitation-related satisfaction was assessed using the SAT-16 (Satisfaction with Acute Treatment-16) questionnaire. The original SAT-16 is a validated scoring system measuring subject satisfaction with respect to the perceived quality of care during inpatient rehabilitation [25]. The questionnaire consisted of a total of 16 4-level items. However, only the questions related to the physicians, physiotherapists, and nursing staff were utilized, which each comprised three items without calculating an overall score. These questions focused on the patient’s relationship with the healthcare professionals, care, and explanations provided. The subjects were given the following response options: dissatisfied, satisfied, or very satisfied. In the absence of a German version of the SAT-16, the questions pertinent to this study were translated into German by the research team.

Additionally, this study inquired about the overall subject’s rehabilitation-related satisfaction using a 5-point Likert scale ranging from 1 (very satisfied) to 5 (very dissatisfied). Further, the progress of various physical ability aspects during the subjects’ stay in the rehabilitation clinic, namely mobility, pain levels, and walking ability was assessed. For each item, the subjects were given the option to indicate whether their situation improved, remained the same, or worsened during the rehabilitation period.

### 2.4. Other Outcome Measurements

The participants’ psychological well-being was assessed using the WHO-5 score at t0 and t1. The WHO-5 is a validated questionnaire that measures well-being and emotional health [26]. It consists of five questions assessing the emotional state over the past two weeks. A total of six response options (e.g., 0 = not feeling well at all, 5 = feeling well all the time) were available, allowing for a maximum score of five points per question and a total score ranging from 0 to 25. Lower scores indicate a lower level of well-being. To obtain a percentage score ranging from 0 to 100, the raw score is multiplied by 4. The data were further categorized into three groups based on the total WHO-5 scores: “severe depressive symptoms” (0–28), “mild-to-moderate depressive symptoms” (29–50), and “no depressive symptoms” (>50).

Furthermore, the Western Ontario and McMaster Universities Osteoarthritis Index (WOMAC)—a validated, commonly used score for TKA subjects—was used to assess the subjects’ functional status before (t0) and after (t1) TKA. The WOMAC consists of 24 questions divided into three subcategories: pain, stiffness, and joint function [27,28]. Each question is rated on a 5-point Likert scale from 0 (no symptoms) to 4 (severe symptoms). These scores are ultimately combined to create a total WOMAC score, which ranges from 0 to 96. A higher WOMAC score indicates a greater disease burden or worse functional status in the subject. 

The subjects’ generic health-related QoL was assessed at t0 and t1 using the EuroQol-Visual Analogue Scale (EQ-VAS) [29]. By means of this validated instrument, the subjects rated their current health status on a scale from 0 to 100, with higher values indicating better health. Relevant demographic information such as age, sex, and body mass index (BMI) at t0 were extracted from the subjects’ electronic health records. The level of education was classified according to the International Standard Classification of Education (ISCED) [30].

### 2.5. Data Analysis

Descriptive analyses of the sample characteristics incorporated percentages and/or mean and standard deviation (SD) stratified by sex. The level of rehabilitation-related satisfaction and its changes over the course of the rehabilitation program were also analyzed descriptively. The subjects were categorized as “satisfied” if they reported being “very satisfied” or “satisfied” with the rehabilitation process. The second group comprised subjects who expressed being “partially satisfied”, “dissatisfied”, or “very dissatisfied” with the rehabilitation. Additionally, 95% confidence intervals (CI) and *p*-values for the responses were calculated, and the results were presented separately for each sex.

A univariable logistic regression was conducted to evaluate the characteristics associated with higher satisfaction with the rehabilitation process. This included sex (men, women); age group (18–64 years, 65–74 years, 75+ years); BMI (<25 kg/m^2^, 25–<30 kg/m^2^, ≥30 kg/m^2^); smoking status (no [longer], current smoker); level of education (high, middle, low); marital status (married, not married); living alone (yes, no); current use of analgesics (yes, no); WOMAC total score; depressive symptoms (WHO-5 score; severe, mild-to-moderate, no); and satisfaction with the surgical outcome after 2 months (satisfied, not satisfied). Finally, all variables were included in a multivariable model. The odds ratios (OR) were calculated with a 95% CI and were considered statistically significant if the accompanying 95% CI did not include 1. All of the calculations were performed using IBM SPSS Statistics, Version 27 and SAS (Version 9.4, SAS Institute, Cary, NC, USA).

## 3. Results

### 3.1. Response and Baseline Characteristics 

In total, 296 subjects undergoing elective TKA between December 2019 and May 2021 were screened (Figure 1). Six of them were excluded due to language barriers and one subject was excluded due to cognitive inabilities. Additionally, six of the subjects had already participated with the contralateral side. Thus, 283 subjects met the study’s inclusion criteria, and 241 subjects consented to participate, resulting in a response of 85.2% in t0. Out of the 241 enrolled subjects, 236 participated in t1, resulting in a response of 97.9%.

Overall, 59.3% of the study subjects were female (Table 1). On average, the subjects were 68.2 years old, 64.0% of participants had a BMI of ≥ 30 kg/m^2^ (mean: 32.7 kg/m^2^), 67.5% were married, and 18.6% had a high education level. The average WOMAC score was 50.5 points, and the average EQ-VAS score was 54.0. The general state of health was reported as moderate to (very) poor in 68.4% of the cases. Overall, 71.5% of the subjects used pain medication preoperatively. On average, the subjects were hospitalized for 9.3 days. Over two thirds of the subjects (69.4%) expressed being “satisfied” or “very satisfied” with the surgical outcome two months post-TKA. There were no substantial differences between men and women regarding age and BMI. Women were more likely to be living alone (29.0% vs. 13.0%) and being unmarried (39.9% vs. 21.9%). Men had higher education levels (24.0% vs. 14.0%). Women were more likely to experience depressive symptoms (64.9% vs. 51.0%) and to have stayed slightly longer in the hospital than men (9.4 vs. 9.1 days).

### 3.2. Utilization of Rehabilitation

In total, out of the 226 subjects who provided information regarding their post-operative rehabilitation usage, 94.7% underwent rehabilitation measures, with inpatient rehabilitation being the predominant choice for 85.4% (Table 1). Overall, the subjects spent on average, 21.5 and 25.4 days in inpatient and outpatient rehabilitation, respectively. Women were more likely to attend inpatient rehabilitation (87.1% vs. 83.0%). There were no sex-related differences in rehabilitation duration. The average time until inpatient rehabilitation after discharge was 3.1 days (n = 175) (Table A1). More than two thirds (64.0%) of all subjects were directly transferred from the acute hospital to inpatient rehabilitation (Table A2). An additional 22.9% started their inpatient rehabilitation within the first week after discharge. In contrast, the average time to start outpatient rehabilitation (n = 19) following discharge was significantly longer (i.e., 11.7 days on average). Most of the subjects (57.9%), however, began their outpatient rehabilitation within the first seven days after discharge (see Table A3).

### 3.3. Symptoms and Functional Change During Rehabilitation

On average, all evaluated aspects improved during rehabilitation (Table 2). Pain at rest improved in 68.4% of the subjects. Moreover, movement-associated pain improved in 77.5% of the subjects. Knee flexion and extension improved in 78.7% and 76.3% of the subjects, respectively. Walking with an aid improved in 78.3% of the subjects, while walking without an aid improved in only 58.9%. 

Men, on average, achieved greater improvements in most evaluated aspects compared to women. Women experienced an exacerbation of pain at rest (9.1% vs. 3.5%) and pain during movement (8.1% vs. 2.4%) more frequently than men. There were no sex-specific differences observed for knee flexion and extension changes. Men experienced more improvement in stair-walking ability (65.1% vs. 57.5%). Moreover, a higher percentage of men improved their walking without aids compared to women (63.9% vs. 55.3%). Women slightly tended to experience worsened wound healing. Overall, in 14.0% of cases, at least one parameter worsened in men, while it was 19.4% for women.

### 3.4. Rehabilitation-Related Subject Satisfaction

Most subjects (77.6%) were satisfied or very satisfied with their rehabilitation in general, with women being more satisfied than men (80.6% vs. 73.3%) (Table 3). The lowest satisfaction scores were observed regarding satisfaction with their physicians. Only 34.0% of the subjects were very satisfied with the medical care provided by the treating physicians, 35.8% with the physician–subject relationship, and 34.3% with the explanations provided by the treating physicians. Men were noticeably less satisfied compared to women as 24.4% of men were dissatisfied with at least one of the aspects (i.e., medical care, subject–provider relationship, and explanations), while, for women, it was only 14.0%. The subjects expressed the highest level of satisfaction with their physiotherapists. In total, the subjects that were very satisfied were 53.8% with physiotherapeutic care, 57.7% with the physiotherapist–subject relationship, and 59.0% with explanations given by their treating physiotherapists. Again, male subjects were less satisfied (14.0% of men were dissatisfied with at least one of the abovementioned aspects vs. 5.5% dissatisfied women). Satisfaction with the nursing staff ranged in the middle, with most of the subjects being at least satisfied. Once again, more women tended to be more satisfied compared to men (17.4% of men were dissatisfied with at least of the aspects vs. 7.8% dissatisfied women).

### 3.5. Factors Associated with Rehabilitation-Related Subject Satisfaction After TKA

Univariable logistic regression models showed that satisfaction with surgery after two months was associated with rehabilitation-related satisfaction (OR 3.41) and the WOMAC score (OR 0.98) (Table 4). Furthermore, it indicated a tendency for women to be more satisfied with the rehabilitation compared to men (OR 1.52). However, this finding did not reach statistical significance. Multivariable logistic regression analysis revealed that female sex (OR 2.51) and being satisfied with surgery after two months (OR 4.50) were associated with a higher probability of being satisfied with the rehabilitation, while WOMAC score was no longer statistically significant.

## 4. Discussion

This study found that 94.7% of TKA subjects, predominantly in an inpatient setting, underwent structured postoperative rehabilitation. Overall, 77.6% of the subjects expressed being satisfied or very satisfied with their rehabilitation measures. Male subjects exhibited greater progress in terms of pain relief and walking ability during their rehabilitation process. Despite this, they remained less satisfied with the rehabilitation measures. The highest levels of satisfaction were reported for interactions with physiotherapists, while the lowest satisfaction was associated with physician interactions.

### 4.1. Utilization of Rehabilitation

The finding that 94.7% of the subjects attended TKA rehabilitation, with the majority being inpatient (85.4%), aligns with the common practice in the German rehabilitation system [10,19,21]. Given this study’s high numbers of rehabilitation utilization, it emphasizes a significant contrast to most international paradigms. According to different authors, rehabilitation after TKA is uncommon in England, Wales, Northern Ireland, and the Isle of Man [15,16]. Similarly, only 34.3% of the subjects attended any form of rehabilitation following TKA in the USA, and it was even less in Canada with only 7.7% [16]. According to another study from the USA, which included both TKA and THA, 45.6% attended a specialized facility, with the majority being skilled nursing facilities [31]. In countries other than Germany, outpatient physiotherapy following TKA was much more frequently utilized, often replacing the need for inpatient rehabilitation. In Japan, on the other hand, the utilization of inpatient rehabilitation services seemed similarly high. According to a recent study from October 2023, which included 51.332 patients from 3033 hospitals, 94% of patients undergoing THA utilized inpatient rehabilitation. The average duration, at 47 days, was higher than in Germany [32].

A similar picture is seen in other medical indications. A study from Germany, which focused on THA subjects, reported comparable overall rehabilitation attendance (92.0%), with about 20% of the subjects deciding on outpatient rehabilitation [22]. As another example, cardiac rehabilitation has been traditionally conducted in an inpatient setting in Germany [33] in contrast to all other European countries and the USA, where outpatient rehabilitation is predominantly conducted [34,35]. Moreover, in Germany, inpatient rehabilitation is considerably more commonly used after disc surgery, where only 36.6% of the patients perform outpatient rehabilitation [36]. In contrast, a study from the USA involving 6921 subjects who underwent disc surgery only recorded a proportion of 9.4% for inpatient rehabilitation [37].

### 4.2. Rehabilitation-Related Satisfaction 

Overall, most of the subjects (77.6%) expressed being satisfied with the rehabilitation program. For comparison, a study with TKA subjects from the USA reported that 76.0% of subjects expressed being satisfied [38]. Although rehabilitation system organizations vary considerably between countries, most subjects report high rehabilitation-related satisfaction. As previously discussed, due to the limited use of inpatient rehabilitation internationally, studies often focus on alternative forms of therapy after TKA. Naylor et al. conducted a comparative analysis of group-based versus one-on-one physiotherapy after TKA [39]. In alignment with the results of this study, overall satisfaction attained a high level (75%), and there were no significant differences in satisfaction between the different therapy deliveries. Moffet et al. compared home visits with telerehabilitation after TKA, where the subjects’ high satisfaction levels (85%) did not differ between the two groups [40]. It appears that intensive measures such as one-on-one care and home visits do not necessarily result in higher levels of patient satisfaction. No differences in rehabilitation-related satisfaction levels between inpatient and outpatient subjects were found in the current study. However, the number of patients who performed outpatient rehabilitation during this investigation was rather small, which precludes a sound conclusion. Based on the current literature, however, it can be speculated that, for subjects following TKA, the overall improvement is the most important aspect, where the specific approach to an optimal post-TKA rehabilitation may not be as important.

### 4.3. Satisfaction with the Rehabilitation Service Providers

This study suggests that rehabilitation-related subject satisfaction was highest for physiotherapist-provided processes. Subjects spend a considerable amount of time during their rehabilitation with these healthcare professionals. In German inpatient rehabilitation centers, a physician usually conducts the initial assessment and performs regular ward rounds. However, it is often the physiotherapist who is the patient’s first point of contact during rehabilitation, which is why patients are potentially the most satisfied with this group of healthcare professionals. Selected authors investigated satisfaction levels in TKA patients undergoing various rehabilitation measures. They attributed satisfaction to adequate time spent with therapists, no change in therapists, and a low number of other subjects during therapy sessions [39]. It is, therefore, plausible that satisfaction with the physicians was lower in comparison. The subjects appeared to require more time, attention, and in-depth information from their physicians. A lack of information for subjects can lead to reduced compliance and poorer well-being [41]. Furthermore, it is recognized that physicians’ communication can create a placebo effect [42]. Heightened medical attention can positively influence subjects’ well-being and satisfaction. Moreover, it is understood that physicians’ behavior and communication can influence self-reported patient health [43]. Improved communication between patients, physiotherapists, and physicians, for instance through interdisciplinary rounds inspired by similar practices in other fields [44], may constitute a viable strategy for boosting patient satisfaction with their physicians. Also, increased direct communication between physicians and patients could potentially lead to improved satisfaction.

### 4.4. Factors Associated with Rehabilitation-Related Subject Satisfaction After TKA

Multivariate regression analyses revealed statistically significant higher rehabilitation-related satisfaction in women versus men. This is surprising considering this study’s findings indicate men achieve better progress during rehabilitation. Few relevant differences in baseline criteria may explain this: the WOMAC and EQ-VAS scores, as well as the general state of health, were lower in the female subjects. This aligns with previous research [45] and suggests that men underwent surgery with a better health status, while women with a poorer health status prior to surgery may experience greater benefits. One noticeable difference is that, prior to surgery, a higher proportion of women took pain medication (80.7%) compared to men (58.1%). Previous research has established that pain relief is a crucial factor for subject satisfaction after TKA [46], and it also highlights that women experience more pain [37]. Additionally, a study found that pain scores were significantly and inversely associated with rehabilitation-related satisfaction [47]. Rehabilitation may have been more satisfying for women due to their comparatively poorer health conditions prior to intervention compared to their male counterparts. It is also possible that women may delay surgery out of fear [48]. Furthermore, it is known that men are offered surgical interventions faster and are more likely to undergo the procedure. Notably, the satisfaction with surgery after two months was almost the same between men and women (69.5% vs. 69.3%). This underlines the result that the lower rehabilitation-related satisfaction of men was probably not caused by a higher dissatisfaction with the surgery itself. A study, comprising 217 subjects who had undergone TKA, showed that, although women had an inferior preoperative functional status than men, there were no significant differences in outcomes at 6 and 24 months. In other words, while initial results may vary, they tend to converge over time, indicating an underlying trend [49]. It was further shown that patients who were satisfied with the surgery itself after two months were also statistically significantly more likely to be satisfied with their rehabilitation. Satisfaction with the surgical outcome appears to be an important factor in patient satisfaction with the subsequent follow-up care. Other potentially influencing factors did not show statistically significant results. Thus, satisfaction with rehabilitation seems to be largely independent of factors such as age, BMI, functional knee status, or depressive symptoms. 

Future research should investigate the most effective components of rehabilitation programs. Moreover, when considering international concepts, innovative outpatient approaches could be used to achieve high rehabilitation-related patient satisfaction. To enhance satisfaction, it may be necessary to address individual needs using a more personalized approach, focusing on comprehensive assessments and tailored interventions to more specifically target functional problems [50].

### 4.5. Strengths and Limitations

A major advantage of this study is that the subjects were directly recruited before surgery, permitting a highly precise evaluation of their preoperative condition. Additionally, the first follow-up conducted two months post-surgery serves as another strong point of this study. While this study’s follow-up interval differs from that of most other studies in the field, making any direct comparison challenging, most of the subjects had just completed their rehabilitation measures at that time, allowing for a highly precise estimation of their health and satisfaction. In addition, a response of 85.2% at t0 and even 97.9% at t1 indicated a representative sample of the overall population of the treated subjects in the hospital.

However, some limitations should also be considered. Firstly, data were collected at one hospital only. Although this hospital serves subjects from a vast catchment area (where merely 31% of subjects reside in the city where the hospital is located), the results may have limited generalizability to other regions and hospitals in Germany. Nonetheless, the subjects had access to multiple rehabilitation centers, leading to data from various facilities being included. Unfortunately, no information regarding the rehabilitation clinics visited or the specific organization of individual therapy sessions can be provided. Consequently, the exact nature of the treatments received cannot be presented. However, the evidence-based rehabilitation standards for TKA patients, as defined by the German Pension Insurance, equally apply to all specialized orthopedic rehabilitation clinics in Germany. Therefore, it can be assumed that any differences in rehabilitation would not have had a substantial impact on the results. Secondly, because of the COVID-19 pandemic, there were two recruitment stops due to the cancellation of elective surgeries (between 17 March and 13 May 2020 and between 18 December 2020 and 31 January 2021). Consequently, the subjects who underwent surgery later than initially planned faced significantly extended waiting times than was originally anticipated. Therefore, it is possible that a diverse subject population was involved, resulting in different outcomes in relation to surgery satisfaction and, subsequently, rehabilitation. In a prior analysis of this study, it was demonstrated that subjects who underwent surgery after the recruitment stops obtained substantially more physiotherapy. However, this occurrence only affected a modest number of 51 subjects [19]. It is worth mentioning that the COVID-19 pandemic might have impacted the subjects’ utilization and their satisfaction with rehabilitation services. Although the data indicate that most of the subjects were able to attend a rehabilitation facility, four of the subjects explicitly mentioned a lack of access to rehabilitation, citing COVID-19 restrictions as a contributing factor. Furthermore, the pandemic may have imposed restrictions within the rehabilitation facilities, potentially impacting the subjects’ overall experience and satisfaction. An important consideration is that the extent of improvement through rehabilitation is based on self-reports rather than being based on validated objective tests. However, the use of patient-reported outcome measures aimed precisely to present the personal opinions of the subjects. Furthermore, 5.3% of the patients did not utilize rehabilitation. This could be due to various reasons, which we unfortunately did not capture. It is also worth mentioning that the patients’ physical ability and functional status were not measured at the initiation of rehabilitation. Lastly, a detailed investigation of the differences between inpatient and outpatient subjects would have been interesting. Unfortunately, the number of outpatient subjects was too small to allow for such analysis. Future research on this aspect is required. 

## 5. Conclusions

This study identified a high utilization of rehabilitation services after TKA and the overall predominantly high rehabilitation-related satisfaction. The highest levels of satisfaction were reported for interactions with physiotherapists, and the lowest were associated with physician interactions. Statistically significant dissatisfaction was found among men and subjects who were overall dissatisfied with the surgery itself already. Future research should investigate which are the most effective components during TKA rehabilitation, having in mind that achieving high subject satisfaction may also be accomplished through innovative outpatient approaches.

## Figures and Tables

**Figure 1 healthcare-12-02099-f001:**
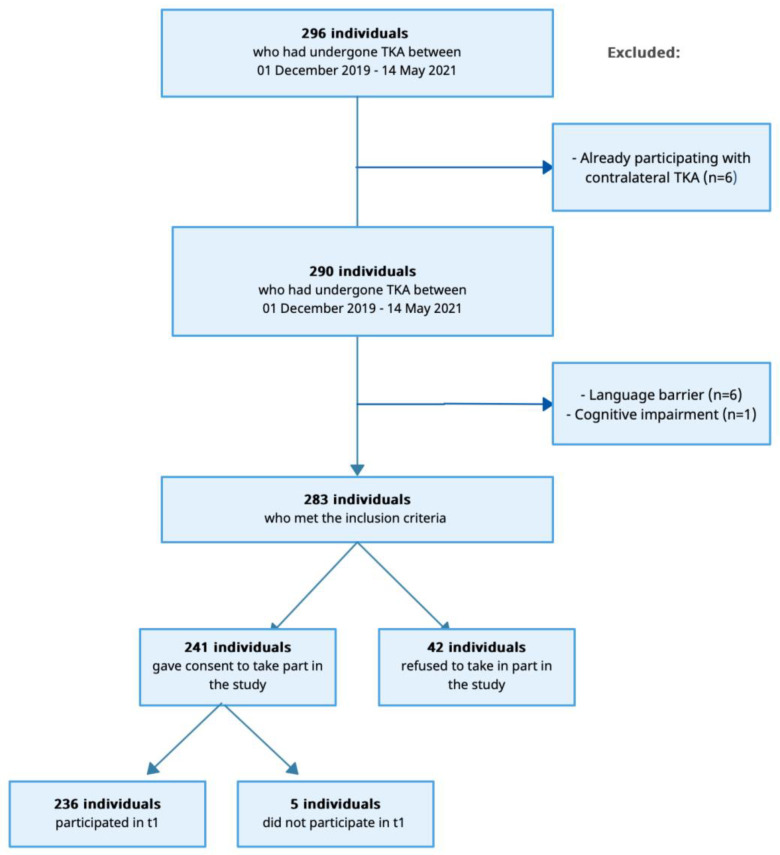
Flowchart of the study population. TKA = total knee arthroplasty.

**Table 1 healthcare-12-02099-t001:** Characteristics of the study population in %.

Characteristics	Total (100%; n = 236)	Male (40.7%; n = 96)	Female (59.3%; n = 140)
Age in years, mean (SD) (n = 236), t0	68.2 (9.4)	68.5 (9.1)	68.0 (9.6)
18–64 years	35.6 (84)	32.3 (31)	37.9 (53)
65–74 years	36.4 (86)	39.6 (38)	34.3 (48)
75+ years	28.0 (66)	28.1 (27)	27.9 (39)
BMI, mean (SD) (n = 236), t0	32.7 (5.9)	32.5 (5.1)	32.8 (6.4)
<25	7.6 (18)	6.3 (6)	8.6 (12)
25–<30	28.4 (67)	26.0 (25)	30.0 (42)
≥30	64.0 (151)	67.7 (65)	61.4 (86)
Living alone (n = 230), t0	22.6 (52)	13.0 (12)	29.0 (40)
General state of health (n = 234), t0			
Good/very good	31.6 (74)	35.4 (34)	29.0 (40)
Moderate	45.7 (107)	40.6 (39)	49.3 (68)
Poor/very poor	22.7 (53)	24.0 (23)	21.7 (30)
Marital status (n = 234), t0			
Not married	32.5 (76)	21.9 (21)	39.9 (55)
Married	67.5 (158)	78.1 (75)	60.1 (83)
Level of education (n = 232), t0			
High	18.1 (42)	24.0 (23)	14.0 (19)
Middle	31.9 (74)	25.0 (24)	36.8 (50)
Low	50.0 (116)	51.0 (49)	49.3 (67)
Current use of analgesics (n = 228), t0	71.5 (163)	58.1 (54)	80.7 (109)
Smoking status (n = 235), t0			
No (longer)	90.2 (212)	90.5 (86)	90.0 (126)
Current smoker	9.8 (23)	9.5 (9)	10.0 (14)
WOMAC score, mean (SD) (n = 225), t0	50.5 (14.5)	47.1 (14.6)	53.0 (14.0)
EQ-VAS (n = 226), mean (SD), t0	54.0 (21.0)	56.8 (21.4)	52.3 (20.7)
Depressive symptoms (WHO-5) (n = 228), t0			
None	40.8 (93)	48.9 (46)	35.1 (47)
Mild-to-moderate	23.7 (54)	22.3 (21)	24.6 (33)
Severe	35.5 (81)	28.7 (27)	40.3 (54)
Length of hospital stay in days, mean (SD) (n = 236)	9.3 (3.0)	9.1 (2.8)	9.4 (3.1)
Satisfaction with surgery after 2 months (N = 232)			
Satisfied/very satisfied	69.4 (161)	69.5 (66)	69.3 (95)
Partly dissatisfied/dissatisfied/very dissatisfied	30.6 (71)	30.5 (29)	30.7 (42)
Utilization of rehabilitation (n = 226)			
Inpatient	85.4 (193)	83.0 (78)	87.1 (115)
Outpatient	9.3 (21)	9.6 (9)	9.1 (12)
None	5.3 (12)	7.4 (7)	3.8 (5)
Duration of inpatient rehabilitation in days, mean (SD) (N = 182)	21.5 (5.7)	21.8 (5.9)	21.3 (5.6)
Duration of outpatient rehabilitation in days, mean (SD) (N = 19)	25.4 (11.6)	24.6 (13.3)	26.1 (10.6)

Values are presented as the mean ± SD for continuous characteristics and as percentages otherwise. SD = standard deviation; n = number of subjects; BMI = body mass index in kg/m^2^; WOMAC = Western Ontario and McMaster Universities Osteoarthritis Index; EQ-VAS = EuroQol-visual analogue scale; and WHO-5 = WHO-Five Well-Being Index. Unless otherwise specified, the values refer to t1.

**Table 2 healthcare-12-02099-t002:** Development during the rehabilitation in % (CI).

Characteristics	Total	Male	Female	*p*-Value
Development pain at rest (n = 206)				0.193
Worsened	6.8 (3.3–10.3)	3.5 (0.0–7.5)	9.1 (3.9–14.3)	
About equal	24.8 (18.8–30.7)	22.4 (13.4–31.3)	26.4 (18.5–34.4)	
Improved	68.4 (62.0–74.8)	74.1 (64.7–83.5)	64.5 (55.7–73.0)	
Development pain during movement (n = 209)				0.152
Worsened	5.7 (2.6–8.9)	2.4 (0.0–5.6)	8.1 (3.2–12.9)	
About equal	16.7 (11.6–21.9)	20.0 (11.4–28.6)	14.5 (8.3–20.8)	
Improved	77.5 (71.8–83.2)	77.6 (68.7–86.6)	77.4 (70.0–84.8)	
Development flexion of the knee (n = 211)				0.896
Worsened	4.3 (1.5–7.0)	3.5 (0.0–7.4)	4.8 (1.0–8.6)	
About equal	17.1 (11.9–22.2)	17.4 (9.4–25.5)	16.8 (10.2–23.4)	
Improved	78.7 (73.1–84.2)	79.1 (70.4–87.7)	78.4 (71.1–85.7)	
Development extension of the knee (n = 211)				0.941
Worsened	3.8 (1.2–6.4)	3.5 (0.0–7.4)	4.0 (0.5–7.5)	
About equal	19.9 (14.5–25.3)	20.9 (12.3–29.6)	19.2 (12.2–26.2)	
Improved	76.3 (70.5–82.1)	75.6 (66.4–84.7)	76.8 (69.3–84.3)	
Development of stair walking (n = 203)				0.449
Worsened	5.4 (2.3–8.6)	3.6 (0.0–7.7)	6.7 (2.3–11.2)	
About equal	34.0 (27.4–40.6)	31.3 (21.3–41.4)	35.8 (27.2–44.5)	
Improved	60.6 (53.8–67.4)	65.1 (54.7–75.4)	57.5 (48.6–66.4)	
Development of safe walking without aids (n = 197)				0.478
Worsened	6.6 (3.1–10.1)	6.0 (0.9–11.2)	7.0 (2.3–11.7)	
About equal	34.5 (27.8–41.2)	30.1 (20.2–40.1)	37.7 (28.7–46.7)	
Improved	58.9 (52.0–65.8)	63.9 (53.4–74.3)	55.3 (46.1–64.5)	
Development of safe walking with aids (n = 198)				0.912
Worsened	2.5 (0.3–4.7)	2.5 (0.0–6.0)	2.5 (0.0–5.4)	
About equal	19.2 (13.7–24.7)	17.7 (9.2–26.2)	20.2 (12.9–27.4)	
Improved	78.3 (72.5–84.1)	79.7 (70.8–88.7)	77.3 (69.7–84.9)	
Development of wound healing (n = 211)				0.509
Worsened	5.2 (2.2–8.2)	4.7 (0.2–9.1)	5.6 (1.5–9.7)	
About equal	10.0 (5.9–14.0)	12.8 (5.7–19.9)	8.0 (3.2–12.8)	
Improved	84.8 (80.0–89.7)	82.6 (74.5–90.6)	86.4 (80.3–92.5)	
At least one parameter worsened during rehabilitation (n = 215)				0.302
At least one worsened	17.2 (12.1–22.3)	14.0 (6.6–21.3)	19.4 (12.5–26.3)	
None worsened	82.8 (77.7–87.9)	86.0 (78.7–93.4)	80.6 (73.7–87.5)	

n = number of subjects; CI = confidence intervals. All values are taken from the t1 questionnaire.

**Table 3 healthcare-12-02099-t003:** Satisfaction with the rehabilitation service providers in % (CI).

Characteristics	Total	Male	Female	*p*-Value
Physicians				
Medical care (n = 215)				0.153
Dissatisfied	10.2 (6.1–14.3)	15.1 (7.5–22.7)	7.0 (2.5–11.4)	
Satisfied	55.8 (49.1–62.5)	53.5 (42.9–64.1)	57.4 (48.8–66.0)	
Very satisfied	34.0 (27.6–40.3)	31.4 (21.5–41.3)	35.7 (27.3–44.0)	
Relationship with physicians (n = 212)				0.217
Dissatisfied	11.8 (7.4–16.2)	16.5 (8.5–24.4)	8.7 (3.7–13.6)	
Satisfied	52.4 (45.6–59.1)	50.6 (29.9–61.3)	53.5 (44.8–62.3)	
Very satisfied	35.8 (29.3–42.4)	32.9 (22.9–43.0)	37.8 (29.3–46.3)	
Physicians’ explanations (n = 213)				0.280
Dissatisfied	15.0 (10.2–19.9)	19.8 (11.3–28.3)	11.8 (6.2–17.5)	
Satisfied	50.7 (44.0–57.5)	47.7 (37.0–58.3)	52.8 (44.0–61.5)	
Very satisfied	34.3 (27.8–40.7)	32.6 (22.6–42.5)	35.4 (27.0–43.8)	
Dissatisfied with at least one: physicians (n = 215)				0.051
dissatisfied with at least one	18.1 (12.9–23.3)	24.4 (15.3–33.6)	14.0 (7.9–20.0)	
Not dissatisfied	81.9 (76.7–87.1)	75.6 (66.4–84.7)	86.0 (80.0–92.1)	
Physiotherapists				
Physiotherapeutic care (n = 212)				0.009
Dissatisfied	7.5 (4.0–11.1)	14.0 (6.6–21.3)	3.2 (0.1–6.3)	
Satisfied	38.7 (32.1–45.3)	39.5 (29.1–50.0)	38.1 (29.5–46.6)	
Very satisfied	53.8 (47.0–60.5)	46.5 (35.9–57.1)	58.7 (50.1–67.4)	
Relationship with physiotherapists (n = 213)				0.708
Dissatisfied	3.8 (1.2–6.3)	3.5 (0.0–7.4)	3.9 (0.5–7.3)	
Satisfied	38.5 (31.9–45.1)	41.9 (31.3–52.4)	36.2 (27.8–44.6)	
Very satisfied	57.7 (51.0–64.4)	54.7 (44.0–65.3)	59.8 (51.2–68.4)	
Physiotherapist explanations (n = 212)				0.961
Dissatisfied	4.2 (1.5–7.0)	4.7 (0.2–9.1)	4.0 (0.5–7.4)	
Satisfied	36.8 (30.3–43.3)	37.2 (26.9–47.5)	36.5 (28.0–45.0)	
Very satisfied	59.0 (52.3–65.6)	58.1 (47.6–68.7)	59.5 (50.9–68.2)	
Dissatisfied with at least one: physiotherapists (n = 214)				0.032
Dissatisfied with at least one	8.9 (5.0–12.7)	14.0 (6.6–21.3)	5.5 (1.5–9.4)	
Not dissatisfied	91.1 (87.3–95.0)	86.0 (78.7–93.4)	94.5 (90.6–98.5)	
Nursing				
Nursing care (n = 211)				0.503
Dissatisfied	5.7 (2.5–8.8)	5.8 (0.8–10.8)	5.6 (1.5–9.7)	
Satisfied	51.2 (44.4–58.0)	55.8 (45.2–66.4)	48.0 (39.2–56.8)	
Very satisfied	43.1 (36.4–49.9)	38.4 (28.0–48.7)	46.4 (37.6–55.2)	
Relationship with nursing (n = 213)				0.402
Dissatisfied	7.0 (3.6–10.5)	8.1 (2.3–14.0)	6.3 (2.0–10.6)	
Satisfied	47.9 (41.1–54.7)	52.3 (41.7–63.0)	44.9 (36.2–53.6)	
Very satisfied	45.1 (38.3–51.8)	39.5 (29.1–50.0)	48.8 (40.0–57.6)	
Nursing explanations (n = 210)				0.739
Dissatisfied	6.7 (3.3–10.1)	8.1 (2.3–14.0)	5.6 (1.5–9.7)	
Satisfied	54.3 (47.5–61.1)	54.7 (44.0–65.3)	54.0 (45.2–62.9)	
Very satisfied	39.0 (32.4–45.7)	37.2 (26.9–47.5)	40.3 (31.6–49.0)	
Dissatisfied with at least one: nursing (n = 214)				0.032
Dissatisfied with at least one	11.7 (7.3–16.0)	17.4 (9.4–25.5)	7.8 (3.1–12.5)	
Not dissatisfied	88.3 (84.0–92.6)	82.6 (74.5–90.6)	92.2 (87.5–96.9)	
Overall rehabilitation-related satisfaction (n = 210)				0.344
Dissatisfied/very dissatisfied	7.6 (4.0–11.2)	10.5 (3.9–17.0)	5.6 (1.5–9.7)	
Partly/partly	14.8 (9.9–19.6)	16.3 (8.4–24.1)	13.7 (7.6–19.8)	
Satisfied/very satisfied	77.6 (71.9–83.3)	73.3 (63.8–82.7)	80.6 (73.6–87.7)	

n = number of subjects; CI = confidence intervals. All values are taken from the t1 questionnaire.

**Table 4 healthcare-12-02099-t004:** Factors associated with rehabilitation-related satisfaction: results from univariable and multivariable logistic regression analyses (n = 182).

Characteristics	Reference	Univariable Analysis OR; (95% CI)	*p*-Value	Multivariable Analysis OR; (95% CI)	*p*-Value
Sex (n = 210)			0.208		0.037
women	Men	1.52 (0.79–2.92)		**2.51 (1.06–6.00)**	
Age group (n = 210)			0.171		0.446
18–64	75+	0.42 (0.17–1.04)		0.52 (0.18–1.50)	
65–74	75+	0.58 (0.23–1.46)		0.54 (0.18–1.67)	
BMI (n = 210)			0.417		0.810
<25	≥30	2.48 (0.54–11.38)		1.53 (0.27–8.77)	
25–<30	≥30	1.35 (0.63–2.90)		0.85 (0.34–2.13)	
Smoking status (n = 209)			0.539		0.593
No (longer)	Current smoker	1.50 (0.41–5.40)		1.54 (0.32–7.40)	
Level of education (n = 207)			0.753		0.689
Middle	Low	0.77 (0.37–1.59)		0.78 (0.32–1.89)	
High	Low	1.01 (0.39–2.62)		1.27 (0.42–3.89)	
Marital status (n = 209)			0.247		0.190
Married	Not married	0.65 (0.31–1.35)		0.23 (0.03–2.07)	
Living alone (n = 205)			0.313		0.326
Yes	No	1.54 (0.66–3.59)		0.32 (0.03–3.16)	
Current use of analgesics t0 (n = 204)			0.231		0.633
Yes	No	0.61 (0.27–1.37)		0.77 (0.26–2.25)	
WOMAC score total (n = 202)		0.98 (0.95–1.00)	0.042	0.97 (0.94–1.01)	0.135
Depressive symptoms (WHO-5) (n = 203)			0.685		0.927
Mild-to-moderate	No	0.74 (0.31–1.75)		0.86 (0.29–2.54)	
Severe	No	0.73 (0.34–1.58)		1.05 (0.38–2.89)	
Satisfaction with surgery after 2 months (N = 207)			0.0004		0.0004
Satisfied	Not satisfied	3.41 (1.73–6.75)		**4.23 (1.89–9.45)**	

The odds ratios of variables significantly associated with satisfaction with rehabilitation are shown in bold. n = number of subjects; TKA = total knee arthroplasty; OR = odds ratio; CI = confidence limit; BMI = body mass index in kg/m^2^; WOMAC = Western Ontario and McMaster Universities Osteoarthritis Index; and WHO-5 = WHO-Five Well-Being Index. Unless otherwise specified, the values refer to t1.

## Data Availability

The datasets generated during and/or analyzed during the current study are available from the corresponding authors on reasonable request.

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
