# Peer review of "Utilization of Rehabilitation Services and Rehabilitation-Related Patient Satisfaction Following Total Knee Arthroplasty—Results of the Prospective FInGK Study"

_healthcare, 2024, doi:10.3390/healthcare12212099_

Round 1
Reviewer 1 Report
Comments and Suggestions for Authors
Dear authors, I had the opportunity to review your manuscript, and I hope my comments are helpful.
In general, was the current study part of what was published before (https://www.ncbi.nlm.nih.gov/pmc/articles/PMC9630069/)?, if so, please indicate the reason for this possible “salami slicing”.
ABSTRACT:
-Please do not repeat “this study” at the start of each paragraph.
INTRODUCTION:
-Please indicate the aim more clearly; it is better to report it as a primary objective and see if there are further secondary objectives.
METHODS:
-Please explain the reason behind selecting 2 and 12 months to collect the satisfaction data.
-did the authors exclude the immediate postoperative in-hospital rehab, as this was provided for all patients?
-Line 118, ref. “20” is not for the SAT-16, please correct.
-Was the SAT-16 translated into German?
RESULTS:
-Please do not repeat data reported in tables in the text.
-Please, in table 1 indicate the time of scores reporting.
-Line 192, the authors reported “no difference.” how was this difference calculated? What was the p-value?
-Line 196, “slightly longer” was this statistically significant or not?
-selecting rehabilitation place (in-hospital or outside) was selected based on which parameters? Patient, surgeon or insurance company decision.
-In most of the tables, the authors did not report any statistically significant testing; they just reported numbers and percentages.
-why the authors did not report the data based on satisfaction or based on the place of rehabilitation instead of reporting based on gender (table 3).
Author Response
Dear Reviewer,
Thank you very much for your helpful comments and suggestions for improvement. To provide better clarity, we have compiled all responses in a detailed response letter.
Best regards

Reviewer 2 Report
Comments and Suggestions for Authors
Dear Authors,
Reading the title, the work suggests a long-term assessment of patients. However, when we read the methodology, it is a QoL assessment after two months. Treatment, or rather postoperative management, is difficult to assess after such a short time, for example, the condition of the scar, healing of tissue, reproducing receptors and creating a new map of neurons or neuromuscular improvement, which can sometimes last 6-12 months.
Detailed suggestions below:
Introduction
Line 52 - if this is the case, it is worth providing the source of such a statement that rehabilitation significantly affects QoL.
Line 54-55 - Rapid mobilization - did the authors mean verticalization or independence of the patient. THIS is not clear to the reader.
Line 55 - Additionally, Rapid mobilization does not depend on the time of commencement of rehabilitation or physiotherapy. There are many factors influencing, e.g. circulatory and respiratory condition. It is worth correcting. Line 57 How is chronic pain relieved? Is this probably achieved by TKA? Maybe it's about postoperative pain?
Line 57-58 What dysfunction do the authors mean?
Line 59-61 The main goals in rehabilitation or rather in physiotherapy have probably changed over the course of 10-15 years. It is worth using newer studies.
Line 63-73 The entire paragraph does not contribute anything to the subject of the work. If after each comparison the authors provide the effect of such treatment on QoL, then this is important for this work, otherwise it is unnecessary text. It is simply a presentation of a different health care system.
Methodology
There is a lack of information about the type of endoprosthesis used and the surgical approach. These are significant variables for assessing QoL.
Line 93 Why is a person with poor language skills excluded and, for example, a mute person not? It is worth explaining
Results
If the authors assess patient satisfaction and state in the methodology that the assessment was conducted before and after the TKA procedure (1.2 months), it is worth presenting such results in a table.
In order to determine the validity of the rehabilitation used, it is also worth presenting what exactly was done. It may turn out that one of the treatment centers has a procedure that can improve the condition of the patients.
Discussion
item 4.1 - the entire paragraph does not contribute much to the work. How does it relate to the topic of satisfaction. Do the authors assess satisfaction with the stay in the inpatient ward or satisfaction after TKA? The treatment system is related to the insurance system adopted in a given country. In order to determine whether the length of stay in the inpatient ward has an impact on the functional condition of patients, a separate study should be conducted. However, this work does not concern such a study. Therefore, the entire chapter does not fit into the content of the discussion.
item 4.2 It is worth explaining what the patients were satisfied with. At this point, the authors describe satisfaction with the system, which is different. It is worth specifying exactly what the work is supposed to be about. Whether it is about satisfaction with the healthcare system in a given country or the topic of the manuscript.
point 4.3 did the authors intend to present patients' satisfaction with the procedures used or with the service provided by physiotherapists, i.e. with knowledge or manual skills? Or maybe with the satisfaction of the presence of physiotherapists as people who are on the front line with the patient.
point 4.4
A description of the results of satisfaction is presented rather than a discussion related to gender after rehabilitation than with satisfaction after TKA.
Author Response

(The authors gave the same response as above.)

Reviewer 3 Report
Comments and Suggestions for Authors
Authors conducted a single-center prospective cohort study of adult patients receiving TKA in Germany and executed a cross-sectional analysis on the rehab services use and factors associated with rehab-related satisfaction at 2 months postoperatively. They found that 85.4% of those consented to participate in the study received inpatient rehab, and 77.6% of those receiving rehab were satisfied or very satisfied with their received rehab services. Multivariable analyses showed that sex and satisfaction with the surgery in general at 2 months were associated with rehab-related satisfaction.
Authors provided the much-needed data to address the knowledge gap on the use of and satisfaction with rehab services in German patients with TKA, and the results were informative. However, I recognized some points below that authors shall consider further.
In the Introduction, although the knowledge gap is stated i.e. information is missing re: TKA rehab use and rehab-related satisfaction in Germany, the clinical and public health implications based on these findings are not stated and therefore shall be explained.
Further, there are multiple patient characteristics that could be associated with rehab use patterns and/or patient’s satisfaction level with the received rehab, while these characteristics were not assessed or reported in the manuscript.
- Disease history and duration of osteoarthritis, as well as severity of osteoarthritis before TKA
- Patients’ physical ability and functional status measured at the initiation of rehab
- Patients with osteoarthritis could have chronic comorbidities including cardiovascular diseases, stroke, rheumatic diseases, musculoskeletal diseases, liver disease, irritable bowel syndrome and gastrointestinal bleeds. Comorbidities and the management of comorbidities during rehab treatment period are likely associated with patient’s satisfaction with rehab services and their overall health. However, a broad range of comorbidities and the management of comorbidities/comedication use was not considered.
- When assessing the characteristics associated with higher satisfaction with the rehab process, results of rehab setting (i.e. inpatient rehab and outpatient rehab) was not presented. However, it’s possible that patients eligible for intensive inpatient rehab ended up receiving outpatient rehab as alternatives, and these patients’ functional status might not have recovered to the level they would be satisfied with. In the Discussion, authors mentioned “no differences in rehab-related satisfaction levels between inpatient and outpatient subjects were found in the current study”, while results were not presented. It’ll be essential to add the results in the main text.
- As authors acknowledged in the Limitation, the use patterns of healthcare services including rehab could have been impacted during the covid pandemic time. Patients’ satisfaction with rehab services may have also been affected given the overall impact of pandemic. It’ll be beneficial to include the study time period e.g. pre-/post-covid as a key characteristic.
Minor observations:
- Line 166 – In this study, “level of education” was categorized into high, middle, and low levels. Detailed definition for the three categories will help readers understand the highest educational level that patients achieved.
- Line 177 – Results section mentioned that “One subject was excluded due to cognitive inabilities.” While this exclusion criterium was not brought up in the Methods exclusion criteria section (Lines 91-93).
- Line 280 – In the Discussion on the Utilization of rehabilitation, authors should discuss the reasons for not receiving any rehab in 5.3% of the patients if they collected the corresponding data.

The overall quality of used English language is satisfactory.
Author Response

(The authors gave the same response as above.)

Reviewer 4 Report
Comments and Suggestions for Authors
Overall, this manuscript presents a well-designed and comprehensive study on the utilisation of rehabilitation services and patient satisfaction following total knee arthroplasty (TKA). The use of a prospective cohort approach and the high response rate contribute to the robustness of the findings. The study offers valuable insights into the rehabilitation practices in Germany and highlights key factors associated with patient satisfaction.
However, there are areas where the manuscript could be strengthened. The discussion would benefit from a deeper analysis of international differences in rehabilitation practices and their impact on outcomes. Additionally, the lower satisfaction scores with physicians compared to physiotherapists warrant further exploration, with suggestions for potential improvements. The study's generalisability could be addressed by discussing the representativeness of the hospital setting. Moreover, the impact of the COVID-19 pandemic on the study outcomes deserves a more detailed examination.
Finally, the decision to focus on specific aspects of the SAT-16 questionnaire without an overall score should be justified, and the findings related to wound healing need further discussion. Addressing these points would enhance the clarity and impact of the manuscript, making it more informative for readers and stakeholders in the field.
Comments:
1. The comparison of rehabilitation practices between Germany and countries like the USA, Canada, and Japan is a good start. However, the discussion could benefit from a deeper exploration of why these differences exist and how they might impact patient outcomes.
2. The manuscript notes that satisfaction with physicians is lower compared to physiotherapists. It would be helpful to delve into the reasons behind this difference and suggest ways to improve patient satisfaction with their physicians.
3. Since the data were collected from a single hospital, the generalisability of the findings may be limited. It would be useful to discuss how representative this hospital is of other settings in Germany and whether similar results could be expected in different regions or healthcare systems.
4. The authors touch on the impact of the COVID-19 pandemic on the study results. A more detailed analysis of how the pandemic specifically affected outcomes, particularly in relation to surgery and rehabilitation delays, would strengthen the discussion.
5. The SAT-16 questionnaire is used to measure satisfaction with acute treatment, but only physician, physiotherapist, and nursing staff interactions are considered, without calculating an overall score. It might be worth explaining the rationale behind this choice and discussing whether an overall score would have provided additional insights.
6. The study reports that wound healing worsened in 5.2% of subjects, but this is not discussed in detail. It would be valuable to explore this finding further, including potential reasons for delayed or poor wound healing and its effect on overall satisfaction and rehabilitation outcomes.
Comments on the Quality of English Language
Some sentences in the manuscript are long and complex. Shortening and simplifying sentences where possible could improve readability.
Author Response

(The authors gave the same response as above.)

Round 2
Reviewer 2 Report
Comments and Suggestions for Authors
Dear Autors,
Thank you for accepting my suggestions and for making the changes and including them in the manuscript.